# Clinical prediction rule for SARS-CoV-2 infection from 116 U.S. emergency departments 2-22-2021

**Jeffrey A. Kline**[1]*, **Carlos A. Camargo, Jr.**[2], **D. Mark Courtney**[3], **Christopher Kabrhel**[2], **Kristen E. Nordenholz**[4], **Thomas Aufderheide**[5], **Joshua J. Baugh**[2], **David G. Beiser**[6], **Christopher L. Bennett**[7], **Joseph Bledsoe**[8], **Edward Castillo**[9], **Makini Chisolm-Straker**[10], **Elizabeth M. Goldberg**[11], **Hans House**[12], **Stacey House**[13], **Timothy Jang**[14], **Stephen C. Lim**[15], **Troy E. Madsen**[16], **Danielle M. McCarthy**[17], **Andrew Meltzer**[18], **Stephen Moore**[19], **Craig Newgard**[20], **Justine Pagenhardt**[21], **Katherine L. Pettit**[1], **Michael S. Pulia**[22], **Michael A. Puskarich**[23], **Lauren T. Southerland**[24], **Scott Sparks**[25], **Danielle Turner-Lawrence**[26], **Marie Vrablik**[27], **Alfred Wang**[1], **Anthony J. Weekes**[28], **Lauren Westafer**[29], **John Wilburn**[30]

1 Department of Emergency Medicine, Indiana University School of Medicine, Indianapolis, Indiana, United States of America, 2 Department of Emergency Medicine, Massachusetts General Hospital, Harvard Medical School, Boston, Massachusetts, United States of America, 3 Department of Emergency Medicine, University of Texas Southwestern, Dallas, Texas, United States of America, 4 Department of Emergency Medicine, University of Colorado School of Medicine, Aurora, Colorado, United States of America, 5 Department of Emergency Medicine, Medical College of Wisconsin, Milwaukee, Wisconsin, United States of America, 6 Section of Emergency Medicine, University of Chicago, Chicago, Illinois, United States of America, 7 Department of Emergency Medicine, Stanford University School of Medicine, Palo Alto, California, United States of America, 8 Department of Emergency Medicine, Healthcare Delivery Institute, Intermountain Healthcare, Salt Lake City, Utah, United States of America, 9 Department of Emergency Medicine, University of California, San Diego, California, United States of America, 10 Department of Emergency Medicine, Mt. Sinai School of Medicine, New York, New York, United States of America, 11 Department of Emergency Medicine, Warren Alpert Medical School of Brown University, Providence, Rhode Island, United States of America, 12 Department of Emergency Medicine, University of Iowa School of Medicine, Iowa City, Iowa, United States of America, 13 Department of Emergency Medicine, Washington University School of Medicine, St. Louise, Missouri, United States of America, 14 Department of Emergency Medicine, David Geffen School of Medicine at UCLA, Los Angeles, California, United States of America, 15 University Medical Center New Orleans, Louisiana State University School of Medicine, New Orleans, Louisiana, United States of America, 16 Division of Emergency Medicine, Department Surgery, University of Utah School of Medicine, Salt Lake City, Utah, United States of America, 17 Department of Emergency Medicine, Feinberg School of Medicine, Northwestern University, Chicago, Illinois, United States of America, 18 Department of Emergency Medicine, George Washington University School of Medicine, Washington D.C., DC, United States of America, 19 Department of Emergency Medicine, Penn State Milton S. Hershey Medical Center, Hershey, Pennsylvania, United States of America, 20 Department of Emergency Medicine, Oregon Health and Science University, Portland, Oregon, United States of America, 21 Department of Emergency Medicine, West Virginia University School of Medicine, Morgantown, West Virginia, United States of America, 22 Department of Emergency Medicine, University of Wisconsin School of Medicine and Public Health, Madison, Wisconsin, United States of America, 23 Department of Emergency Medicine, Hennepin County Medical Center and the University of Minnesota, Minneapolis, Minnesota, United States of America, 24 Department of Emergency Medicine, Ohio State University Medical Center, Columbus, Ohio, United States of America, 25 Department of Emergency Medicine, Riverside Regional Medical Center, Newport News, Virginia, United States of America, 26 Department of Emergency Medicine, Beaumont Health, Royal Oak, Michigan, United States of America, 27 Department of Emergency Medicine, University of Washington School of Medicine, Seattle, Washington, United States of America, 28 Department of Emergency Medicine, Carolinas Medical Center at Atrium Health, Charlotte, North Carolina, United States of America, 29 Department of Emergency Medicine, Baystate Health, Springfield, Massachusetts, United States of America, 30 Department of Emergency Medicine, Wayne State University School of Medicine, Detroit, Michigan, United States of America

* jefkline@iu.edu



**Data Availability Statement:** Data are within the protocol in the Supporting information files.

**Funding:** The author(s) received no specific funding for this work.

**Competing interests:** The authors have declared that no competing interests exist.

# Abstract

## Objectives

Accurate and reliable criteria to rapidly estimate the probability of infection with the novel coronavirus-2 that causes the severe acute respiratory syndrome (SARS-CoV-2) and associated disease (COVID-19) remain an urgent unmet need, especially in emergency care. The objective was to derive and validate a clinical prediction score for SARS-CoV-2 infection that uses simple criteria widely available at the point of care.

## Methods

Data came from the registry data from the national REgistry of suspected COVID-19 in EmeRgency care (RECOVER network) comprising 116 hospitals from 25 states in the US. Clinical variables and 30-day outcomes were abstracted from medical records of 19,850 emergency department (ED) patients tested for SARS-CoV-2. The criterion standard for diagnosis of SARS-CoV-2 required a positive molecular test from a swabbed sample or positive antibody testing within 30 days. The prediction score was derived from a 50% random sample (n = 9,925) using unadjusted analysis of 107 candidate variables as a screening step, followed by stepwise forward logistic regression on 72 variables.

## Results

Multivariable regression yielded a 13-variable score, which was simplified to a 13-point score: +1 point each for age>50 years, measured temperature>37.5˚C, oxygen saturation<95%, Black race, Hispanic or Latino ethnicity, household contact with known or suspected COVID-19, patient reported history of dry cough, anosmia/dysgeusia, myalgias or fever; and -1 point each for White race, no direct contact with infected person, or smoking. In the validation sample (n = 9,975), the probability from logistic regression score produced an area under the receiver operating characteristic curve of 0.80 (95% CI: 0.79–0.81), and this level of accuracy was retained across patients enrolled from the early spring to summer of 2020. In the simplified score, a score of zero produced a sensitivity of 95.6% (94.8–96.3%), specificity of 20.0% (19.0–21.0%), negative likelihood ratio of 0.22 (0.19–0.26). Increasing points on the simplified score predicted higher probability of infection (e.g., >75% probability with +5 or more points).

## Conclusion

Criteria that are available at the point of care can accurately predict the probability of SARS-CoV-2 infection. These criteria could assist with decisions about isolation and testing at high throughput checkpoints.

## Introduction

The ability to rapidly estimate the probability of infection with the novel coronovirus-2 that causes severe acute respiratory syndrome (SARS-CoV-2) remains a formidable problem. The protean clinical picture of SARS-CoV-2 infection confounds its prediction. For example, the

disease syndrome that SARS-Cov-2 produces—recognized as COVID-19—can manifest a wide range of nasopharyngeal, respiratory, and gastrointestinal symptoms, and a substantial minority of patients who carry SARS-CoV-2 manifest no symptoms at the time of testing [1,2]. Asymptomatic patients can manifest nasopharyngeal viral loads, and shedding capacity similar to symptomatic, infected persons [3,4]. Factors limiting our current knowledge include the lack of systematically collected data from a large, unbiased, geographically diverse samples of patients, and problems associated with limited availability of molecular diagnostic tests and assays, long turnaround time and low diagnostic accuracy [3,5–7]. The need for rapid exclusion without molecular testing arises daily in health care clinics, outpatient treatment facilities, at the point of intake for homeless shelters, judicial centers for incarceration, and extended care facilities. This need is particularly urgent in the emergency department (ED), which represents the largest interface between the general public and unscheduled medical care. In 2016, the >5000 US EDs had approximately 145 million patients [8]. Additionally, because the ED interconnects with both outpatient and inpatient medical care, the critical question of SARS-CoV-2 infection status affects decisions to admit or discharge the patient, return to work, need for home isolation, and the location of hospital admission. These questions become more complicated for patients without access to basic medical care, and those experiencing serious mental illness, substance use disorders, and homelessness.

To address these needs, the authors created the REgistry of suspected COVID-19 in EmeRgency care (RECOVER), a national network to capture data from patients tested for SARS-CoV-2 and evaluated in the ED [9]. This report addresses the primary goal of the initial network-wide registry, which was to create a quantitative pretest probability scoring system (putatively named the COVID-19 Rule Out Criteria score [CORC score])to predict the probability of a SARS-CoV-2 test, with special attention to identify those at very low probability of infection. The intent of the score was to function similarly to the Wells pretest probability scoring criteria and Pulmonary Embolism Rule out Criteria (PERC rule) for acute pulmonary embolism, respectively, except the diagnostic target was SARS-CoV-2 [10,11].

## Materials and methods

The RECOVERY network has resulted from the collaboration of 45 emergency medicine clinician-investigators from unique medical centers in 27 US states. Most of the 45 medical centers participating are the flagships of hospital networks that include community and academic centers. Information about these sites, and the methods of the initial registry, are available elsewhere [9]. The primary objective of the registry was to obtain a large sample of ED patients with suspected SARS-CoV-2 and who had a molecular test performed in the ED as part of their usual care. The design, collection, recording and analysis of data for this report were conducted in accordance with the transparency in reporting of a multivariable prediction model for individual diagnosis and prognosis (TRIPOD) criteria [12]. The RECOVER registry protocol was reviewed by the institutional review boards (IRBs) at all sites; 42 IRBs provided an exemption from human subjects designation, whereas three IRBs provided approval with waiver of informed consent. All data were anonymized prior to analysis.

Briefly, eligibility for enrollment required that a molecular diagnostic test was ordered and performed in the ED setting with suspicion of possible SARS-CoV-2 infection, or COVID-19 disease [9]. Patients could only be enrolled once. Otherwise, there were no age or symptom-based exclusions; however, the guidance was provided to exclude patients where the test was clearly done for automated, administrative purposes in the absence of any clinical suspicion for infection. One example for exclusion was patients without suspected infection but who had swab testing performed in the ED done only to comply with a hospital screening policy for

admissions or pre-operative testing. All sites were contracted to abstract charts from at least 500 patients.

Data were collected from the electronic medical record, using a combination of electronic download for routinely collected, coded variables (e.g., age, vital signs and laboratory values), supplemented by chart review by research personnel, using methods previously described [9]. Each REDcap data form included 204 questions resulting in 360 answers, because many questions allowed multiple answers. Data were archived in the REDCap® system, with electronic programming to ensure completion of mandatory fields and sensible ranges for parametric data. Training of data abstractors was done via teleconference with the principal investigator (JAK) and program manager (KLP), supplemented by an extensive guidance document and field notes present in the REDCap® system, visible to the person doing enrollment. This analysis was pre-planned as the first manuscript from the RECOVER network.

To generate a comprehensive pool of independent variables for a quantitative pretest probability model, as well as harmonization with other data, among the 204 questions, we recorded 28 symptoms, including all symptoms from the Clinical Characterization Protocol from the World Health Organization-supported International Severe Acute Respiratory and Emerging Infection Consortium (ISARIC) [13]. The REDCap form also recorded 14 contact exposure risks, ranging from no known exposure, to constant exposure to a household contact with COVID-19. We anticipated that many patients would have multiple ED visits prior to testing, especially for atypical presentations, and the goal was to collect patient data from the earliest medical presentation. Accordingly, the symptoms and contact risks, together with the vital signs (body temperature, heart rate, respiratory rate, systolic and diastolic blood pressure and pulse oximetry reading) were recorded from the first ED visit within the previous 14 days (the "index visit"). The form also documented presence or absence of 18 home medications and 39 questions about past medical history. Outcomes, including results of repeated molecular testing, or antibody testing for SARs-CoV-2 were recorded up to 30 days after the date of the SARS-CoV-2 test that qualified the patient for enrollment.

The criterion standard for disease positive was evidence of SARs-CoV-2 infection, from either a positive molecular diagnostic test from a swab sample (usually from the nasopharynx), or a positive serological IgM or IgG antibody, documented within 30 days of enrollment. The criterion standard for disease negative required that patients have no positive molecular or serological test for SARS-CoV-2 or clinical diagnosis of COVID-19 within 30 days.

## Model development

After upload, each REDCap form was inspected centrally for completeness and sensibility of data, resulting in verification queries. For example, if a patient had none of 28 symptoms, the site investigator was asked to double-check the medical record. Uploaded records were considered eligible for analysis after resolution of queries coupled with electronic verification by the data abstractor and site investigator that each uploaded record was a true and complete reflection of data in the medical record. The *a priori* plan for model development called for a classical approach of screening candidate variables with unadjusted analyses, followed by multivariable logistic regression analysis with conversion into either a scoring system or a set of criteria. The REDCap data collection form was produced in March 2020, when the phenotype of patients with SARs-CoV-2 infection was incompletely understood. Thus, the plan was an agnostic approach: to screen all potential variables for discriminative value, including method and day of arrival to the ED, patient demographics, symptoms, vital signs, contact risks, habits, medications and past medical history. As previously described, we estimated a minimum sample size of 20,000 to allow derivation and validation on approximately 10,000 patients in each

step, assuming a 30% criterion standard positive rate and a greater than 10:1 ratio of outcomes to variables, recognizing this as a minimum criterion [9,12,14].

To derive the CORC score, we first extracted a 50% random sample, and, for statistical testing, used the criterion standard result as the dependent variable. Per protocol, categorical data that were not charted were considered absent, but missing continuous data (>0.1%, age, vital signs, and body mass index) were analyzed for monotonicity and replaced using multiple imputations method in SPSS® (IBM Corp. Released 2020. IBM SPSS Statistics for Windows, Version 27.0. Armonk, NY: IBM Corp). The mean values from five iterations were used. Bivariate data were compared between test + and test—using the Chi-Square statistic and means from parametric variables (e.g., age) were compared using unpaired t-test. Variables with P<0.05 were entered into logistic regression equation, initially leaving parametric data as continuous (to create the probability from logistic regression), and variables selected for score development using an empirical stepwise forward approach using the likelihood ratio approach. "The model was terminated when the change Akaiki information criterion (AIC = 2k-2ln(L) where k = number of variables and L = maximum likelihood) reached its nadir. Model fitness was assessed with the Hosmer-Lemeshow test.

To produce the actual CORC score (a simplified version of the logistic regression equation), we dichotomized continuous data at the midpoint of the difference in means between patients with and without SARS-CoV-2 infection. To test for validity, the probability from logistic regression was computed by solving for probability (P) from the logistic regression equation (obtained from the antilog of the logistic regression equation yielding P = [1+exp(-Σcoefficients+intercept)]$^{-1}$); the net positive points for the CORC score were calculated for each of the remaining 50% of patients in the registry, who were independent of the derivation population. Diagnostic accuracy of the probabilityfrom logistic regression and CORC score were assessed in the validation with receiver operating characteristic curve and diagnostic indexes from contingency table analysis. Data were analyzed with SPSS® software with the Complex Sampling and Testing module.

## Results

Data for this analysis were downloaded from the registry on December 3, 2020. The download included 20,060 complete records collected per protocol from 41 hospital systems representing 116 unique hospitals from 25 states. Eligible records came from patients tested for SARS-CoV-2 from the first week of February, until the fifth week of October, 2020. After exclusion of 210 records marked by the sites as screen fails (from a later discovered exclusion criterion), 19,850 records were left for analysis. Multiple imputation successfully replaced all missing parametric values, including body mass index as the most frequently missing value (in 25%), followed by respiratory rate (1.4%). For age, blood pressure, and pulse oximetry, values were missing in less than 1% of the samples. Each record was then assigned a random number drawn from 1 to 19,850, and re-sequenced, and the first and second halves were used to derive and test the probabilityfrom logistic regression, respectively. Table 1 shows the clinical characteristics of the sample, divided into the derivation and validation groups, and indicates that random sampling produced two comparable groups. Compared with US Census Bureau data from 2019, the median age of this ED sample is older by approximately 12 years, and has approximately a 13% higher representation of persons identifying as Black (and lower percentage of persons identifying as White), but a similar distribution of biological sex and Hispanic or Latino ethnicity [15]. Table 1 conveys findings that are important to developing accurate pretest probability criteria using criteria available at the bedside. First, the pooled prevalence of infection among those tested was 34%, which is relatively high for producing exclusionary criteria.

**Table 1. Clinical features of the derivation and validation samples.**

| | Derivation (n = 9925) | | Validation (N = 9925) | |
|---|---|---|---|---|
| | mean | SD | mean | SD |
| Age (years) | 50 | 20.6 | 51 | 20.4 |
| Number of symptoms at presentation | 4.5 | 1.9 | 4.6 | 1.9 |
| Duration of symptoms (days) | 5.4 | 9.7 | 5.5 | 10.1 |
| Heart rate (beats/min) | 94.7 | 21.6 | 94.7 | 21.3 |
| Respiratory rate (breaths/min) | 20.1 | 5.5 | 20.0 | 5.4 |
| Pulse oximetry at triage (%) | 96 | 6.0 | 96 | 6.5 |
| Lowest pulse oximetry reading (%) | 94 | 7.3 | 94 | 7.6 |
| Temperature (Celsius) | 37.1 | 1.1 | 37.1 | 1.2 |
| Systolic blood pressure (mm Hg) | 134 | 25.6 | 134 | 25.3 |
| Diastolic blood pressure (mm Hg) | 80 | 16.6 | 80 | 16.4 |
| Body mass index (Kg/m^2) | 30 | 10.1 | 30 | 9.8 |
| Days between SARS-CoV-2 test order and result | 1.4 | 2.3 | 1.5 | 2.4 |
| | n | % of group | n | % of group |
| Age<18 years | 425.0 | 4% | 444.0 | 4% |
| Female sex | 5201 | 52% | 5226 | 53% |
| Asian race | 289 | 3% | 252 | 3% |
| Black race | 2630 | 26% | 2686 | 27% |
| White race | 5301 | 53% | 5180 | 52% |
| Hispanic or Latino ethnicity | 1748 | 18% | 1729 | 17% |
| Homeless | 334 | 3% | 325 | 3% |
| Obese | 2501 | 25% | 2587 | 26% |
| Diabetes mellitus | 2262 | 23% | 2317 | 23% |
| Hyperlipidemia | 2782 | 28% | 2808 | 28% |
| Hypertension | 4157 | 42% | 4229 | 43% |
| Active cancer | 1214 | 12% | 1217 | 12% |
| Prior organ transplantation | 167 | 2% | 178 | 2% |
| Atrial fibrillation | 779 | 8% | 803 | 8% |
| Ischemic heart disease | 950 | 10% | 996 | 10% |
| Heart failure | 968 | 10% | 1007 | 10% |
| Chronic obstructive pulmonary disease | 1027 | 10% | 1019 | 10% |
| Asthma | 1623 | 16% | 1558 | 16% |
| Prior venous thromboembolism | 598 | 6% | 583 | 6% |
| Current smoker | 1792 | 18% | 1770 | 18% |
| No symptoms | 20 | 0% | 13 | 0% |
| SARS-CoV-2 infection | 3443 | 35% | 3422 | 34% |
| Influenza testing done* | 4514 | 45% | 4506 | 45% |
| Other viral testing† | 3873 | 39% | 3839 | 39% |
| Chest radiograph done | 7499 | 76% | 7493 | 75% |
| Laboratory analysis of blood specimen | 7600 | 77% | 7624 | 77% |

*Influenza A, B or both positive = 279/90020 (3% positive rate)

†One or more other viruses detected = 1122/7712 (15% positive rate).

Second, the mean turn-around-time for SARS-CoV-2 testing was greater than one day, although the median time was 0.5 days (interquartile range 0–1.0). Third, approximately 5% of the sample had none of the 28 recorded symptoms at presentation, but still had clinical suspicion that led to testing. Fourth, approximately one-quarter of all patients had no chest

**Table 2. Logistic regression results of the selected model (the probability from logistic regression).**

|  | Coefficient | Odds ratio | 95% CI | |
|---|---|---|---|---|
|  |  |  | Lower | Upper |
| Black race | 0.88 | 2.40 | 2.04 | 2.82 |
| White race | -0.42 | 0.66 | 0.57 | 0.76 |
| Hispanic or Latino ethnicity | 1.34 | 3.81 | 3.28 | 4.42 |
| Age in years | 0.02 | 1.02 | 1.02 | 1.02 |
| Symptom: Loss of sense of taste or smell | 1.93 | 6.89 | 5.22 | 9.11 |
| Symptom: Non-productive cough | 0.43 | 1.54 | 1.39 | 1.70 |
| Symptom: Fever | 0.44 | 1.55 | 1.39 | 1.72 |
| Symptom: Muscle aches | 0.48 | 1.61 | 1.43 | 1.81 |
| Exposure to COVID-19: None known | -0.45 | 0.64 | 0.57 | 0.71 |
| Exposure to COVID-19: Household contact with known or suspected infection | 1.68 | 5.36 | 4.42 | 6.51 |
| Pulse oximetry at triage | -0.04 | 0.96 | 0.95 | 0.97 |
| Temperature in Celsius | 0.44 | 1.55 | 1.46 | 1.65 |
| Current smoker | -0.81 | 0.45 | 0.39 | 0.51 |
| Intercept | -14.78 | N/A |  |  |

Model analysis: Hosmer Lemeshow P = 0.526, McFadden's pseudo R2 = 0.22, C statistic = 0.80.

radiograph performed and almost one-quarter had no laboratory analysis of a blood specimen. Additional data of relevance include the fact that 1,915 patients (10%) had visited the ED within the previous 14 days prior to testing for SARS-CoV-2, but records documented clinical suspicion for COVID-19 in only 367 (19%) of these visits.

Of 107 candidate variables shown in S1 Table, 72 had P<0.05 by univariate statistical analysis (Chi-Square for bivariate data and unpaired t-test for continuous data), comparing data from patients with positive SARS-CoV-2 testing versus patients with negative test. These 72 variables were subsequently evaluated by stepwise forward multivariable logistic regression using the likelihood ratio method. After exclusion of 42 variables that were not significant, the procedure was repeated with 30 variables. The model selected for the probability from logistic regression was from step 13 (13 variables) based upon consideration of the need to limit number of variables for practical use with maintenance of model fitness by keeping the Hosmer-Lemeshow P value >0.10. These 13 variables were then examined by a single-step logistic regression to produce Table 2. This model produced a C-statistic (area under the receiver operating characteristic curve) of 0.80 (0.79–0.81). When the equation solved for probability, at a cutoff of 0.1 this yielded sensitivity of 97% and specificity of 20% in the derivation population. Table 3 shows the simplification of the probability from logistic regression into the 13 component CORC score, which included the dichotomization of age, temperature and the pulse oximetry reading obtained at the time of triage in the ED. With the exception of the anosmia/dysgeusia variable, the use of whole digits (-1 or +1, score range -3 to +10) proportionately reflect the sign and rounded magnitude of the beta coefficients and intercept obtained from repeated logistic regression with age, pulse oximetry and temperature converted to dichotomous variables with cutoffs at 50 years, 94.5% and 37.5°C respectively. With 0 or fewer points considered a test negative CORC score result, a negative CORC score produced diagnostic sensitivity of 96% and specificity of 21% in the derivation population.

When applied to the other half of the sample (validation group, n = 9925), the probability from logistic regression and CORC score performed similarly. The probability from logistic regression had an area under the receiver operating characteristic of 0.80 (95% CI, 0.79 to

**Table 3. The COVID-19 rule out criteria (CORC score).**

| Component | Points |
|---|---|
| Black race | +1 |
| White race | -1 |
| Hispanic or Latino ethnicity | +1 |
| Age >50 years | +1 |
| Symptom: Loss of sense of taste or smell | +1 |
| Symptom: Non-productive cough | +1 |
| Symptom: Fever | +1 |
| Symptom: Muscle aches | +1 |
| Exposure to COVID-19: None known | -1 |
| Exposure to COVID-19: Household contact with known or suspected infection | +1 |
| Pulse oximetry at triage <95% | +1 |
| Temperature in Celsius >37.5 | +1 |
| Current smoker | -1 |

0.81), and when solved for probability (P), if P<0.1, the diagnostic sensitivity was 96.8% (96.1 to 97.3%) and the specificity was 20.1% (19.1 to 21.1%), yielding a posterior probability of 7.8% (6.5 to 9.8%). The accuracy of the probability from logistic regression in the validation dataset was maintained across the month of diagnosis. For the 8,444 patients evaluated early in the US pandemic (February to May 2020) the area under the receiver operating characteristic curve for the probability from logistic regression was 0.80 (0.79 to 0.81), compared with 0.81 (0.79 to 0.84) for 1,481 patients evaluated from June 2020 onward. The probability from logistic regression area under the receiver operating characteristic curve was decreased in patients with zero symptoms (0.73, 0.69 to 0.77).

In the validation set, the CORC score negative (0 or fewer points from Table 3) produced sensitivity of 95.6% (94.8 to 96.3%), specificity of 20.0% (19.0 to 21.0%), likelihood ratio negative of 0.22 (0.19 to 0.26) and a posterior probability of 10.4% (8.9 to 12.1%). The probability of infection increases with the number of positive points from the CORC score. This stepwise, positive concordance is shown in Fig 1, which plots the posterior probability of positive SARS-CoV-2 testing as a function of the number of points from the CORC score from the validation population. The probability of SARs-CoV-2 infection is >75% in a patient with +5 or more points from the CORC score.

Table 4 shows the standard diagnostic contingency table (also referred to as a confusion matrix) using a CORC score >0 as the definition of a positive test result with associated calculations of precision, recall and F1 index. Fig 2 shows the plots of the precision-recall curve and receiver operating characteristic curve with their AUC data. The CORC score had a slightly lower area under the curve (0.75, 0.74–0.76) than the probability from logistic regression (0.80, 0.79–0.81).

Given the concern about low diagnostic sensitivity for molecular testing on swab samples, a relevant question is how the CORC score performed among patients with an initially negative swab test who had subsequent evidence of SARS-CoV-2 infection within 30 days. From the entire sample (both derivation and validation), the initial swab that qualified the patient for enrollment was negative in 13,159 patients. Of these, 174 (1.1%) subsequently had evidence of SARs-CoV-2 from either a repeated nasopharyngeal swab or positive antibody testing done within 30 days. Among these 174 patients who had a possibly false negative molecular test done on a swab sample, the CORC score was >0 in 87%.

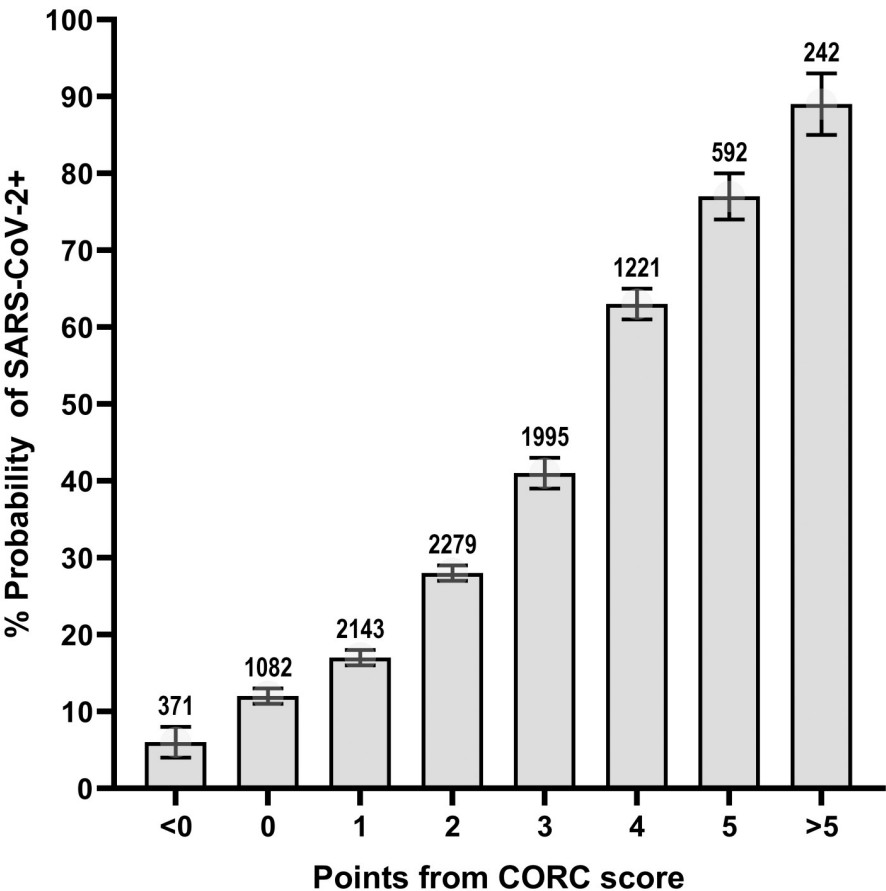

**Fig 1. The CORC score by number of points.** The probability of SARS-CoV-2 infection increased according to number of points from the CORC score.

**Table 4. Contingency table (confusion matrix) for the CORC score.**

|  | CORC>0 | CORC< = 0 |
|---|---|---|
| SARS-CoV-2+ | 3271 | 151 |
| SARS-CoV-2- | 5201 | 1302 |

Precision = 3271/(3272+5201) = 0.39.

Recall = 3271/(3271+151) = 0.96.

F1 = (0.39*0.96)/(0.39+0.96) = 0.55.

Abbreviations: CORC-COVID-19 Rule-Out Criteria.

## Discussion

This work addresses the urgent need for criteria to rapidly, easily, and accurately estimate the probability of SARs-CoV-2 infection. Using registry data from the RECOVER Network, which was specifically created to address this knowledge gap, we found that 13 variables—11 of which were obtained from verbal interview, together with one data point each from a thermometer and a pulse oximeter—can accurately predict the probability of SARS-CoV-2 infection if entered into a logistic regression equation and solved for probability (the probability

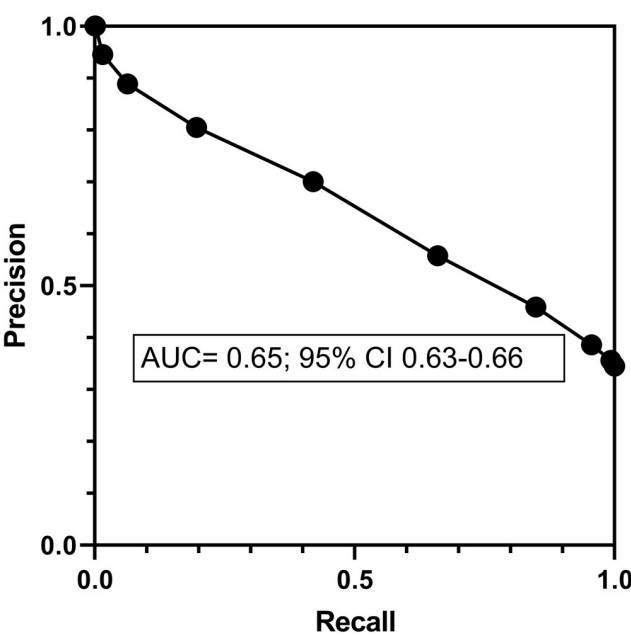

**Fig 2. Diagnostic performance of the CORC score.** The plot in the top panel shows the precision-recall curve and the plot in the lower panel shows the receiver operating characteristic curve for the COVID-19 Rule Out Criteria (CORC) score.

from logistic regression) [9]. A simpler version comprising 13 binary variables, scored with negative or positive point values (the CORC score, shown in Table 3) provides similar accuracy. To our knowledge, this is the first prediction rule (or score) for SARS-CoV-2 infection that does not require laboratory or radiographic data which makes this model very useful in high throughput settings, such as triage areas of emergency departments and also in non hospital settings such as homeless encampments and street medicine.

The probability from logistic regression was derived from a large patient pool enrolled from 27 states with demographics reflective of the overall US population. The overall utility and durability of the probability from logistic regression is suggested by the area under the receiver operating characteristic curve of 0.80 in both the derivation and validation samples, and that this level of accuracy was retained in patients tested either earlier or later in the first year of the US SARS-CoV-2 pandemic. A negative CORC score (0 or fewer points) had 95.6% sensitivity and 20.0% specificity in the validation sample, providing a likelihood ratio negative of 0.22 (95% CI 0.19 to 0.26). Moreover, the CORC score was positive (>0 points) in the 87% of patients with an initially negative and subsequently positive molecular test for SARS-CoV-2 done on a swab sample from the nasopharynx. For the goal of predicting high risk of infection, patients in the validation sample with a +5 or more points from the CORC score had a >75% probability of a positive test. Thus, assuming the expected prevalence of SARS-CoV-2 infection is below 10%, the estimated likelihood ratio negative of 0.22, a CORC score of zero or less would allow a very low posterior probability (e.g., <2.0%) and thus may obviate the need for molecular testing or isolation in a negative pressure room in the ED setting. On the other hand, a CORC score ≥5 should be considered predictive of high risk, suggesting the need for molecular testing, and possibly repeated testing if the first test is negative [16].

In terms of methodological strength, the large, diverse, and representative patient sample of patients tested for SARS-CoV-2 has a low risk of sampling bias, which has hampered previous prediction rules for COVID-19 [17]. The practical benefit of the CORC criteria is the lack of requirement for radiological or laboratory data, which were not ordered in the usual care of over a quarter of patients in this sample and are not available in many settings where risk assessment for probability of SARS-CoV-2 infection is critical to decision-making. These findings suggest that the CORC score, if validated in prospective work, can assist with decisions about need for formal diagnostic testing and isolation procedures for persons passing through high throughput settings including the triage area of some emergency departments and medical clinics, and at the point of entry for homeless shelters, industry, correctional facilities, and extended care facilities. In the home setting thermometers are common and in some protocols pulse oximetry has been used to monitor outpatients with known COVID-19 [18]. Thus, in concept, the CORC score could be an adjunctive measure to assess the probability of SARS-CoV-2 among household contacts of persons known to be infected. After prospective validation, the CORC score may also help reduce low-value repeated molecular testing after initial infection, that could produce false positive results.

The data for the probability from logistic regression and CORC score were obtained retrospectively using rigorous methods to ensure high value variables, and unique, relevant circumstantial data. In contrast to many recent reports using clinical informatics, the level of detail for the data from this study required manual evaluation of medical records by research personnel. For example, manual review was required to ensure that the symptoms recorded represented those that the patient manifested on the first contact with the healthcare system while infected with SARs-CoV-2—which was the case for 1,915, or 10% of the cohort. The probability from logistic regression was a required step to create the simpler CORC score. The more formal probability from logistic regression is calculated by exponentiating the logistic regression equation and solving for probability, a task easily performed using an online or internet-

based calculator. However, recognizing prior literature on the real-world behavior of physicians, we believe a simpler scoring system comprising positive or negative whole single digits will enhance dissemination and adoption [19,20].

The variables retained by the selection process for both the probability from logistic regression and score warrant discussion. First, the large sample and high prevalence would have allowed the stepwise forward logistic regression to retain many more variables, but we terminated the selection at 13 variables for several reasons. The first reason centered on the pragmatic consideration of the time required in busy clinical practice to use the decision aid. The second reason is concern about an overfit model, which is more likely to occur with an excessively complex derivation, regardless of the learning method [21]. Previous simulation studies suggest that the variability of the area under the receiver operating characteristic curve with 13 variables, and a sample size of 10,000 including >30% prevalence of outcomes is less than 5% with repeated sampling [22]. Third, at the 14[th] step, the model began to introduce variables that might be more vulnerable to interobserver variability, including "active cancer" at step 14, a variable that likely requires more inference than the retained 13 variables. A potentially unanticipated finding is that smoking history was retained as a negative predictor of infection —a finding that has been reported by others who have suggested that nicotine may reduce expression of epithelial ACE2 receptor, and thus reduce SARS-CoV-2 infectivity [23–25].

Compared with persons identifying as White, Black race and Latino/Hispanic ethnicity significantly increased the probability of infection. Race-specific patterns in symptom manifestation that might alter clinical suspicion and testing threshold do not appear to explain the differences in positive rate [26]. To our knowledge, no genetic or biologic reasons explain why Black and Hispanic/Latino patients are more likely to have a SARS-CoV-2 infection. Instead, the statistical weight on these variables may result from them acting as proxies for other societal factors. The association of race with positive testing may correlate with a higher likelihood of working service-related jobs which are unable to be done from home (thereby increasing exposure to SARS-CoV-2). In one study of a cohort of SARS-CoV-2 infected patients in Louisiana, 77% of those requiring hospitalization were Black; only 30% of the total area population is Black [27]. However, when adjusted for socioeconomic status and pre-existing clinical comorbidities, there was no racial difference identified in mortality [27]. Ongoing work will report the impact of insurance status and geographic location (by four digit zip code) on SARS-CoV-2 infection rate and severity.

The retrospective collection of data introduces the primary limitation of this work inasmuch as the probability from logistic regression and CORC score performance, including metrics of inter-rater reliability and operational characteristics, have not been used yet in real practice. For example, in terms of generalizability for high throughput screening, it remains unknown whether the temperature component, measured by an infrared thermometer, and the oxygen saturation, measured by a portable pulse oximeter, would provide similar diagnostic accuracy. Symptoms not recorded were assumed to be absent, which could affect score precision and accuracy. Another limitation is the relative lack of data from most recent cases. The most recent patient was evaluated in October and most cases came from early spring of 2020. The genotype of the virus, as well as the phenotype of infected patients, may have changed with time, and the effect on accuracy and imprecision are unknown. Additionally, it remains possible that machine-based learning methods may offer a superior role, although as a preliminary step, several of the authors of this work, directly compared three derivation techniques (logistic regression, random forest and gradient boosting) to create prediction models for SARS-CoV-2 using ED-based data. The logistic regression model had an AUC of 0.89 (95% confidence interval [CI] = 0.84 to 0.94); the random forest method had AUC of 0.86 (95% CI = 0.79 to 0.92) and gradient boosting had an AUC of 0.85 (95% CI = 0.79 to 0.91) [28]. It is important to note that

all prior prediction models included use of laboratory and radiographic values. To consider the possible benefit of machine learning, the authors reviewed the diagnostic accuracy of criteria of 20 reports to predict SARS-CoV-2 infection, including both logistic regression and machine learning techniques [28–30]. This informal scoping review revealed that the diagnostic accuracy of machine learning was not superior to logistic regression-based models, and therefore supported the pre-planned classical approach to model development [28].

In conclusion, we present novel criteria requiring only information that can be obtained from the patient interview, a thermometer, and a pulse oximeter to predict the probability of SARS-CoV-2 infection. A score of zero from the simplified COVID-19 rule-out criteria (the CORC score) predicts a low probability of infection and a score of 5 or more predicts a high probability of infection. If prospectively validated, we believe the CORC score will help expedite decision-making in high throughput settings.

## Supporting information

**S1 Table. P values from unpaired t-test or Chi-Square.**
(DOCX)

**S1 File.**
(PDF)

## Acknowledgments

The authors thank Patti Hogan and Amanda Klimeck for administrative oversight of the RECOVER Network.

## Author Contributions

**Conceptualization:** Jeffrey A. Kline, Carlos A. Camargo, Jr., D. Mark Courtney, Christopher Kabrhel.

**Data curation:** Jeffrey A. Kline, Carlos A. Camargo, Jr., D. Mark Courtney, Thomas Aufderheide, Joshua J. Baugh, David G. Beiser, Christopher L. Bennett, Joseph Bledsoe, Edward Castillo, Makini Chisolm-Straker, Elizabeth M. Goldberg, Hans House, Stacey House, Timothy Jang, Stephen C. Lim, Troy E. Madsen, Danielle M. McCarthy, Andrew Meltzer, Stephen Moore, Craig Newgard, Justine Pagenhardt, Katherine L. Pettit, Michael S. Pulia, Michael A. Puskarich, Lauren T. Southerland, Scott Sparks, Danielle Turner-Lawrence, Marie Vrablik, Alfred Wang, Anthony J. Weekes, Lauren Westafer, John Wilburn.

**Formal analysis:** Jeffrey A. Kline, Carlos A. Camargo, Jr., D. Mark Courtney.

**Funding acquisition:** Jeffrey A. Kline.

**Investigation:** Jeffrey A. Kline, Carlos A. Camargo, Jr., Kristen E. Nordenholz, Thomas Aufderheide, Joshua J. Baugh, David G. Beiser, Christopher L. Bennett, Joseph Bledsoe, Edward Castillo, Makini Chisolm-Straker, Elizabeth M. Goldberg, Hans House, Stacey House, Timothy Jang, Stephen C. Lim, Troy E. Madsen, Danielle M. McCarthy, Andrew Meltzer, Stephen Moore, Craig Newgard, Justine Pagenhardt, Katherine L. Pettit, Michael S. Pulia, Michael A. Puskarich, Lauren T. Southerland, Scott Sparks, Danielle Turner-Lawrence, Marie Vrablik, Alfred Wang, Anthony J. Weekes, Lauren Westafer, John Wilburn.

**Methodology:** Jeffrey A. Kline, Carlos A. Camargo, Jr., D. Mark Courtney, Christopher Kabrhel, Kristen E. Nordenholz.

**Project administration:** Jeffrey A. Kline, Carlos A. Camargo, Jr., D. Mark Courtney, Christopher Kabrhel, Kristen E. Nordenholz.

**Resources:** Jeffrey A. Kline.

**Software:** Jeffrey A. Kline.

**Supervision:** Jeffrey A. Kline, Christopher Kabrhel.

**Validation:** Jeffrey A. Kline.

**Writing – original draft:** Jeffrey A. Kline, Carlos A. Camargo, Jr., D. Mark Courtney, Christopher Kabrhel, Kristen E. Nordenholz.

**Writing – review & editing:** Jeffrey A. Kline, Carlos A. Camargo, Jr., D. Mark Courtney, Christopher Kabrhel, Kristen E. Nordenholz, Thomas Aufderheide, Joshua J. Baugh, David G. Beiser, Christopher L. Bennett, Joseph Bledsoe, Edward Castillo, Makini Chisolm-Straker, Elizabeth M. Goldberg, Hans House, Stacey House, Timothy Jang, Stephen C. Lim, Troy E. Madsen, Danielle M. McCarthy, Andrew Meltzer, Stephen Moore, Craig Newgard, Justine Pagenhardt, Katherine L. Pettit, Michael S. Pulia, Michael A. Puskarich, Lauren T. Southerland, Scott Sparks, Danielle Turner-Lawrence, Marie Vrablik, Alfred Wang, Anthony J. Weekes, Lauren Westafer, John Wilburn.

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
