## [Decision Letter · Decision Letter 0]

18 Feb 2021

PONE-D-21-00785

Clinical prediction rule for SARS-CoV-2 infection from 116 U.S. emergency departments

PLOS ONE

Dear Dr. Kline,

Thank you for submitting your manuscript to PLOS ONE. After careful consideration, we feel that it has merit but does not fully meet PLOS ONE’s publication criteria as it currently stands. Therefore, we invite you to submit a revised version of the manuscript that addresses the points raised during the review process.

We look forward to receiving your revised manuscript.

Kind regards,

Ruslan Kalendar, PhD

Academic Editor

PLOS ONE

2. Please provide additional details regarding participant consent.

Reviewers' comments:

Reviewer's Responses to Questions

**Comments to the Author**

1. Is the manuscript technically sound, and do the data support the conclusions?

Reviewer #1: Yes

Reviewer #2: Partly

Reviewer #3: Yes

2. Has the statistical analysis been performed appropriately and rigorously? 

Reviewer #1: Yes

Reviewer #2: Yes

Reviewer #3: Yes

3. Have the authors made all data underlying the findings in their manuscript fully available?

Reviewer #1: Yes

Reviewer #2: No

Reviewer #3: No

4. Is the manuscript presented in an intelligible fashion and written in standard English?

Reviewer #1: Yes

Reviewer #2: Yes

Reviewer #3: Yes

5. Review Comments to the Author

Reviewer #1: 

Thank you for the opportunity to review this manuscript which describes the development and validation of the (COVID Rule Out Criteria) CORC models for predicting a PCR positive infection with COVID-19 among patients presenting to a large network of US emergency departments. Overall, I am very impressed with this project and identify a number of strengths. Firstly the authors are to be commended on the robustness of the data set as well as the promptness for achieving data collection analysis and reporting within 10 months of the start of the pandemic. The project also benefits from adherance to the tripod reporting guidelines as well as a robust methodology for high quality data collection end a fairly clear approach to the development of both the simplified and more comprehensive CORC rule. Also commendable are the emphasis on readily collected clinical data points that bodes well for this rules uptake into clinical practice. Lastly on the list of kudos as a emergency physician who who has provided care for hundreds of suspected COVID-19 patients the CORC model has significant face validity and jives with what corresponds to both my clinical impression as well as the literature to date.

I do have a few questions and suggestions for the authors to consider. I wonder if an additional justification for the use of the CRC score or rule could relate to the limited performance characteristics of PCR testing. Specifically, if the PCR test have some optimal specificity and can result in false positives from remote and no longer active infections would the CORC be another justification four streamlining testing and not providing it widely in ED settings.

I also wish that the authors can help elaborate on why smoking appears to be protected in the CORC model. Additionally, it is curious to me that dyspnea was not an independent predictor variable and a better explanation of this would be appreciated. Similarly the absence of gastrointestinal symptoms is interest because at the present time the emergency Department where I work will subject patients with those complaints two isolation procedures. I presume therefore that the failure of nausea and vomiting diarrhea to be included in the CORC model is it reflection off it's insignificant contribution to predicting COVID-19 infection.

I think it is important for the authors to provide more clarity on the ethnic associations with COVID-19 infection in order to make the project more generalizable. The key question is whether being black or of Hispanic origin is somehow a genetic predisposition to infection or if from a social determinants of health perspective these groups are more likely to be frontline workers and live in communal settings that places them at risk.

Final point on application of a validated model relates to the potential use of the CORC in a shared decision-making context. Patients present to the emergency Department with an expectation or desire to undergo COVID-19 testing and in theory at least the CRC model can't present an actual risk that can be discussed in the context of the potential downsides of undergoing testing.

In summary this is obviously to me a high impact publication and the models that have been developed should be prioritized for urgent validation. This work should be disseminated promptly so that other groups around the world who are engaged in similar research can have the opportunity to validate the model in their setting.

Reviewer #2: 

Summary

The authors used registry data from the RECOVER network to build a predictor that predicts whether a patient was tested positive or negative for Covid-19. To this end, they used a stepwise forward logistic regression to reduce the number of independent variables to 13. Based on these 13 independent variables, they built a CORC rule that assigns a point system that can be used to decide the likelihood of the patient to have Covid-19. It was found that those with >5 CORC rule (maximum possible score 13) had >75% probability of infection. Although a practical CORC rule that can easily be adapted in the field was given by the authors, the manuscript lacks comparison with other machine learning methods and visual illustration (figures, tables) that helps the reader understand the work.

Major

- Have you tried other machine learning methods? How does the logistic regression compare to the other models like decision tree, SVM, neural network, etc.? The authors state the reason why they chose logistic regression amongst other models in the discussion. But it would be helpful for the readers to show the actual results in the manuscript.

- It’s difficult to understand the landscape of the data used in the manuscript. For example, how many features (independent variables) are there in total? How many of them are categorical variables and how many of them are non-categorical variables?

- How does the logistic regression model differ from the CORC rule and the CORC score? What is the objective of each system?

- The manuscript needs to report the statistical results for different conditions using tables. For example, what is the confusion matrix for the final logistic regression model built using 13 independent variables? It should also include complete statistics like precision, recall, F1, etc. Similarly, figures that show the precision-recall curve and ROC curve should be added.

- How does the performance change as the number of selected features (variables) decreases while performing stepwise logistic regression? And what was the ‘quantitative’ reason for selecting the 13 variables? If we select 15 variables instead of these 13, for example, will we have better performance?

Medium

- I have difficulty understanding the difference between the data obtained through RECOVER (3rd paragraph of the Methods section) and 28 symptoms (4th paragraph of the Methods section). The manuscript makes it think that these are all different sources of data. Also, the authors mention ‘form’. What is this form? It wasn’t clearly defined previously.

- What is the difference between the 30% criterion standard positive rate and the 50% random sampling? Terminologies are not clearly defined here. It would help to have a table that shows how the data was split.

- ‘Variables with P<0.05 were entered into logistic regression equation…’ What are these variables? How do these variables differ from the variables selected for rule development?

- Why is it important to retain the performance of the CORC from the earlier in the year to the later?

- Can you build a better CORC rule by assigning different points to each feature? For example, Assign a higher point for ‘loss of sense of taste or smell’ than the race?

- I could not located the supplemental table 1. The link to the supplemental information directed me to the Kline et al. 'Multicenter registry of United States emergency department

patients tested for SARS-CoV-2'.

Minor

- The authors state in the introduction ‘In 2016, the >5000 US EDs had approximately 145 million patients.’ Are there more recent statistics after the covid-19 pandemic?

- Authors use the term predictor and variable interchangeably. I suggest using ‘variable’ to be consistent and reduce confusion among the readers.

Reviewer #3: 

SUMMARY: The manuscript describes logistic regression-based model that can identify the probability of a positive SARS-CoV-2 test using most accessible clinical features from patients in the US. The manuscript is well written, the data appears to be of high value, the data analysis methodology is appropriate although too simple, and the results are valuable. I have several suggestions that if not addressed can substantially undermine the potential impact of the manuscript.

Note, I did not find the supplemental Table 1.

MAJOR ISSUES

1) Impact of testing method: Given the variable accuracy of SARS-CoV-2 test methods (using swab or blood samples), it is important to analyze and report whether the predictive model is the same, or different regardless of the testing method. Once done, It will be important to emphasize the predictive model that is only trained on a portion of data that uses the more reliable testing method.

2) Comparison with baseline: Authors mention previously published predictive models, however do not show any numerical comparison of such models in the context of their data. For specificity, sensitivity and area under the curve values, the baseline model numbers should also be mentioned (the baseline here would be a model that always predicts the test to be negative).

3) Conclusions are not very actionable: For results to be more actionable, it would be very valuable to provide recommendations in the following form (or something similar, otherwise it seems very unfortunate that such conclusions cannot be made!): Patients with a CORC score above HH.H threshold can be considered COVID-19 positive with 95% confidence and patients with CORC score below LL.L threshold can be considered negative with 95% confidence. For other patients (i.e. CORC score between LL.LL and HH.HH) molecular testing is needed for a high confidence diagnosis. Upon medical resource limitations (timely molecular testing, treatment capacity), the CORC score can be used for testing and treatment prioritization of patients. In areas that molecular testing is unavailable, the prevalence of COVID-19 can be estimated using the average CORC score of patients to be interpreted using Table XX which shows the estimated positivity rate for different CORC scores.

4) Predictive modeling method is too simple: The predictive modelling approach is not wrong but seems too simple. More sophisticated methods such as artificial neural networks (ANN), random forests (RF) or decision tress (DT) that are capable of modeling multiplicative relationships between variables are advisable given that the number of features is low compared to number of records. A DT based model is particularly applicable since it can be directly used to extract rules instead of the point-based formula. I’d suggest to use a data-driven modelling approach (such as DT based) instead of the hand-made model, for providing simple rules since the techniques such as dichotomization are not recommended (see “Royston, P., Altman, D. G., & Sauerbrei, W. (2006). Dichotomizing continuous predictors in multiple regression: a bad idea. Statistics in medicine, 25(1), 127-141.”)

MINOR ISSUES:

1) Authors mention “All sites were contracted to abstract charts from at least 500 patients”. As a person who does not work in a hospital, I don’t understand this sentence.

2) Authors mention “The mean values from five iterations were used, and compared with the pre-imputation mean to confirm a significant change (P<0.05)”: I don’t understand this sentence.

3) Authors mention “variables selected for rule development using an empirical stepwise forward approach using the likelihood ratio approach”, but I didn’t find the exact method and threshold used.

4) Authors mention “values were missing in less than 1% of the sample”. Replace “sample” with “samples”.

5) Missing value imputation creates a potential bias. Therefore will be valuable to also evaluate the model on records that have no missing values for the features that are used by the model.

6) Authors mention “Compared with US Census Bureau data from 2019, this ED sample is older, has a higher representation of persons identifying as Black (and lower percentage of persons identifying as White”. It is valuable to put the relevant numerical values here for the reader to see.

7) “probability (P)” is mentioned a few times without clarifying what it represents/means here. This maybe a first class term in the data analysis software used (SPSS) that readers may not be familiar with.

6. PLOS authors have the option to publish the peer review history of their article (what does this mean?). If published, this will include your full peer review and any attached files.

Reviewer #1: **Yes: **Eddy Lang

Reviewer #2: **Yes: **Jason Youn

Reviewer #3: No

---

## [Author Response · Author response to Decision Letter 0]

24 Feb 2021

Please see attached response letter

---

## [Editor Report · Decision Letter 1]

26 Feb 2021

Clinical prediction rule for SARS-CoV-2 infection from 116 U.S. emergency departments

PONE-D-21-00785R1

Dear Dr. Kline,

We’re pleased to inform you that your manuscript has been judged scientifically suitable for publication and will be formally accepted for publication once it meets all outstanding technical requirements.

Kind regards,

Ruslan Kalendar, PhD

Academic Editor

PLOS ONE

---

## [Editor Report · Acceptance letter]

2 Mar 2021

PONE-D-21-00785R1 

Clinical prediction rule for SARS-CoV-2 infection from 116 U.S. emergency departments 2-22-2021 

Dear Dr. Kline:

I'm pleased to inform you that your manuscript has been deemed suitable for publication in PLOS ONE. Congratulations! Your manuscript is now with our production department. 

Kind regards, 

on behalf of

Prof. Ruslan Kalendar 

Academic Editor

PLOS ONE